# Rethinking Multiple-Instance Learning From Feature Space to Probability Space

**Zhaolong Du[1], Shasha Mao[1],[*] Xuequan Lu[2], Mengnan Qi[1],**
**Yimeng Zhang[1], Jing Gu[1], Licheng Jiao[1]**
[1]Xidian University, [2] The University of Western Australia

## Abstract

Multiple-instance learning (MIL) was initially proposed to identify key instances within a set (bag) of instances when only one bag-level label is provided. Current deep MIL models mostly solve multi-instance problem in feature space. Nevertheless, with the increasing complexity of data, we found this paradigm faces significant risks in representation learning stage, which could lead to algorithm degradation in deep MIL models. We speculate that the degradation issue stems from the persistent drift of instances in feature space during learning. In this paper, we propose a novel Probability-Space MIL network (PSMIL) as a countermeasure. In PSMIL, a self-training alignment strategy is introduced in probability space to cope with the drift problem in feature space, and the alignment target objective is proven mathematically optimal. Furthermore, we reveal that the widely-used attention-based pooling mechanism in current deep MIL models is easily affected by the perturbation in feature space and further introduce an alternative called probability-space attention pooling. It effectively captures the key instance in each bag from feature space to probability space, and further eliminates the impact of selection drift in the pooling stage. To summarize, PSMIL seeks to solve a MIL problem in probability space rather than feature space. Experimental results illustrate that PSMIL could potentially achieve performance close to supervised learning level in complex tasks (gap within 5%), with the incremental alignment in propability space bring more than 19% accuracy improvements for current existing mainstream models in simulated CIFAR datasets. For existing publicly available MIL benchmarks/datasets, attention in probability space also achieves competitive performance to the state-of-the-art deep MIL models. Codes are available at `https://github.com/LMBDA-design/PSAMIL`.

## 1 Introduction

Multiple-Instance Learning (MIL) (Dietterich et al., 1997; Maron & Lozano-Pérez, 1997) was introduced to identify the key instances within a set of instances when only a bag-level label (indicating whether there is any key instance in the set) is available. Originated in the machine learning era, MIL was initially studied as a pure classification algorithm, thus it heavily relies on simple initial inputs. MIL has found wide applications in coarse label learning for images, videos, and texts. With the rise of the deep learning, the deep MIL models are typically formulated as representation learning, followed by classification. However, existing works in MIL have primarily focused on relatively simple data inputs such as raw properties of molecular, text statistical representations (Andrews et al., 2002; Zhou et al., 2009). Even in recent complex applications such as cancer analysis based on Whole Slide Images (WSI) (Bejnordi et al., 2017) and video-based anomaly detection (Sultani et al., 2018), it is still necessary to simplify the instance representations as much as possible before taking them as inputs for the MIL model. To satisfy the conventional requirements of MIL, such complex data instances need to be pre-processed in a certain way (for example, pre-extracting the instances by a heavyweight pre-trained model) to ensure the initial separability as the model input, to guarantee low classification difficulties. Under such condition, the representation learning stage

---

[*]Corresponding author. Email: ssmao@xidian.edu.cn. This work was supported in part by the State Key Program of National Natural Science of China under Grant 62234010.

could then be simplified to a few (or no) fully-connected layers to achieve satisfactory performance, and in this way deep MIL models could be able to focus on classification stage.

In contrast to the simplistic practice commonly adopted in current MIL, many deep-learning based classification tasks pose significant challenges due to the inherent complexity of the data. It is unlikely that a simplified representation extraction process can effectively handle all complex cases. Rather than taking simplified inputs, representation learning of complex data often requires serious consideration in many weakly-supervised tasks (Rolnick et al., 2017; Zhang et al., 2016).

However, introducing a sophisticated feature extractor for complex data instances in an end-to-end MIL model is not a straightforward process: the increased complexity of the feature extractor brings greater capability, but more instances or more accurate label guidance are meanwhile required to ensure the quality of representation learning. While in cases where data is limited or the label guidance is insufficient, such feature extractors are prone to under-fitting, thus failing to learn useful representations. In the absence of instance labels to guide the representation learning for instances, the complex data/feature extractor may easily lead the MIL model to learn irregular or non-discriminative representations, eventually resulting in degradation for the MIL model (an illustration shown in Figure 1). To address this key problem, we argue that the learning process of every instance in a bag deserves more attention.

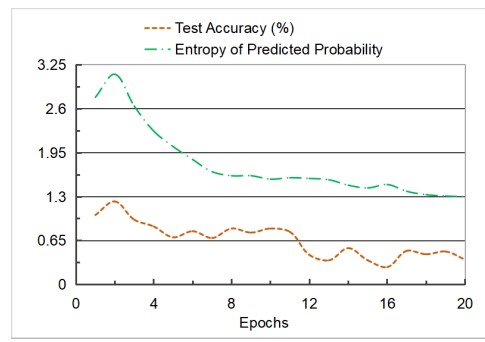

Figure 1: A mainstream MIL model (ABMIL) with complex inputs. For a 100-class classification task, the performance variation over 20 epochs on the CIFAR-100 dataset from ABMIL is reported. The MIL model is trained in multi-instance mode with a complex feature extractor (ResNet) being trained simultaneously.

In this paper, we first point out a rather common issue in current deep MIL models, which is the difficulty in conducting general representation learning on data. As the complexity of data increases, MIL models tend to learn drifted instance representations, ultimately leading to model degradation. To address this, we design and constrain the MIL algorithm in probability space, and propose the Probability-Space MIL network (PSMIL). PSMIL comprises two key strategies implemented in probability space, including probability-space attention pooling and a probability-space alignment objective. To demonstrate the potential degradation issues that current MIL models may face in feature space, we introduce comprehensive simulated datasets to evaluate the ability of MIL models in learning instance representations. On complex tasks, experiments show that the designed probability-space alignment objective effectively constrains instance representations to a more stable space during the representation learning stage, meanwhile bringing non-trivial performance improvements and stability across current MIL methods. In addition, we analyze the mechanism of the widely used attention-pooling method and demonstrate that it is also easily affected by the continuous drift of instance features. Then, the probability-space attention pooling is proposed to further eliminate the impact of drift problem in the pooling stage of MIL paradigm. With these two new strategies in probability space, the model can circumvent the drift issues that traditional MIL models may face in feature space, achieving performance close to that of fully supervised models on more complex tasks (gap within 5%). We also validated our model on various existing MIL datasets with SOTA-level performance.

## 2 RELATED WORK

Multiple-instance learning was initially defined as solving binary classification problems given fixed instance features and bag-level labels, evaluated at the bag-level accuracy during the test. In deep learning era, the mainstream MIL models typically consist of a feature extractor, an instance pooling component, and a classifier concatenated together. The deep MIL model first extracts instance features through the feature extractor, aggregates features into a bag-level feature via the pooling component, and then feeds them into the classifier for classification.

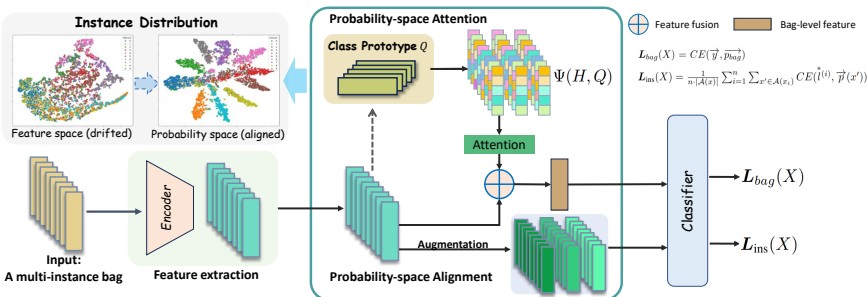

Figure 2: Overview of our Probability-space Multiple-Instance Learning Network (PSMIL).

Research on pooling/classification strategies in deep MIL model is very progressive. Initially, deep MIL models take instances or instance features as input, simply selecting the instance with the highest score (thought as key instance) as the bag-level feature (max pooling). Recently, Attention-based MIL (ABMIL) (Ilse et al., 2018) has been one of the prominent works where the attention module is introduced into deep models as pooling component in the MIL paradigm. These methods have been validated on simple benchmarks such as MNIST images (LeCun et al., 1998) and other manually crafted features (Andrews et al., 2002). In real applications, such as WSI diagnosis/video-based anomaly detection, models generally require well-extracted separable features for input. Besides, there are efforts to extend MIL to multi-class classification problems. Regressor-guided MIL (Du et al., 2024) decomposes $k$-class problems into $k$ binary classification branches, providing solutions on the UNBC pain classification dataset.

Existing works in MIL are usually based on the primitive separability of initial inputs. When the initial inputs are easily ensured separable, the MIL algorithm can mainly focus on pooling components/instance analysis or loss function design. For example, the MNIST dataset, composed of simple gray images, easily ensures separability after the simple feature extraction. The UNBC facial expression pain dataset also shows small intra-class variations where expressions with the same pain-intensity level exhibit remarkable similarity (Lucey et al., 2012). In VAD problem, features pre-extracted by the I3D network as input exhibit better initial separability than C3D, and as a result, more than 5% performance improvement under the same conditions compared to the C3D features was achieved (Kamoona et al., 2023). Similar to the VAD, recent MIL models used in WSI diagnosis also adopt pre-extracted convolutional features as input, including the latest work (Fourkioti et al., 2023).

The premise of high-quality input in the MIL paradigm is reasonable but also oversimplified to some extent. Introducing heavyweight feature extractors required by complex data rather than ensuring high quality separable inputs can easily lead to dilemmas when training both the feature extractor and pooling strategy in two stages. As shown in Figure 1, Attention-based MIL (ABMIL) uses ResNet18 as the feature extractor for multiple-instance learning on CIFAR-100 image bags. During the inference stage, when evaluating the accuracy using single CIFAR image, the classification accuracy at random-guessing level indicates that the model did not learn actual class-relevant instance representation. With the significant model capability brought by a complex feature extractor, the model potentially extracts features in the way that we do not expect. This is what we call the problem of feature drift.

## 3 MATERIAL AND METHODS

### 3.1 PRELIMINARIES: DEEP MIL MODELS REPRESENTED BY ATTENTION-BASED MIL

MIL problem is formulated generally: Given a bag composed of $n$ instances, denoted as $X = [x_1, x_2, \ldots, x_n], X \in \mathbb{R}^{n \times D}$ and a bag label $Y \in \{0, 1\}$ which indicates whether an instance with non-zero label *(key instance)* exists in the bag, we need to learn a model $g : X \rightarrow Y$. The performance of the mapping $g$ is typically demonstrated through bag-level binary classification accuracy and instance localization results, where localization reflects the ability to learn and capture key instances from a bag, forming the inference basis for a MIL model. MIL considers individual instances

to be independent, without any sequential or other correlated relationships. Recently, the concept of MIL has been extended to multi-class problems, and $Y$ is denoted as 0 to $k-1$ in $k$-class scenario.

Due to the effectiveness of the attention mechanism, Attention-based MIL (ABMIL) has been widely applied and further developed. ABMIL and its family are the current mainstream deep MIL models. Deep models in this family often simplify the representation learning stage as much as possible and use attention-like pooling strategy to select key instances to form an overall bag-level feature, which is then passed to a classifier. Here we denote the feature matrix obtained by the feature extractor $Enc$ as $H \in \mathbb{R}^{n \times d}$, where each row of $H$ corresponds to the feature of each instance. The pooling strategy then aggregates the feature matrix $H$ into a bag-level feature $H_{bag} \in \mathbb{R}^d$ to be sent into the linear classifier to get the bag-level prediction $\overrightarrow{p_{bag}}$. The parameters of the final linear classification layer of the MIL model are ($C \in \mathbb{R}^{k \times d}$ for weights, $\overrightarrow{b} \in \mathbb{R}^k$ for biases). When the raw data is relatively simple, it can often be directly used as model input $X$. When the raw data is complex, ABMIL typically requires pre-extracted separable features as input $X$ to reduce the difficulty of representation learning. Through this strategy, ABMIL can perform a form of limited representation learning using several (or no) simplified layers. When setting the feature extractor to 0 layers, $H = Enc(X) = X$. The overall forward process of ABMIL can be stated as follows:

$$H_{bag} = \text{attention\_pooling}(Enc(X)); \quad \overrightarrow{p_{bag}} = \text{softmax}(CH_{bag} + \overrightarrow{b})$$

Recently, there are some advances in attenion-based MIL. Dual-Stream MIL (DSMIL) (Li et al., 2021) adjusts attention-based pooling by introducing an additional max pooling branch. Double-tier feature distillation MIL (DTFDMIL) (Zhang et al., 2022) applies attention-based pooling twice sequentially by dividing the raw bag into several pseudo-bags. Disambiguated attention-Embedding MIL (DEMIL) (Tang et al., 2024) introduces a value mapping operation to transform $H$ into a hidden space $V$ and applies attention-based pooling in hidden feature space to address multi-classification problems. Additionally, there are works that directly use transformer block as the pooling component (TransMIL) (Shao et al., 2021). However, these models all learn with relatively simple feature extractors internally, and thus they are still restricted by the complexity of the data inputs. For general representation learning, there should not be restriction on oversimplifying the feature extractor or data input. We observed that once the restriction is not strictly satisfied, ABMILs tends to degrade to be random guessing. Specifically, the involved major issues are as follows.

***Feature Drift in representation learning.*** Considering to solve a weakly-supervised classification problem, ABMILs needs to obtain discriminative features relevant to the classification task through the feature extractor. Given a commonly-used feature extractor with greater capability, when insufficient guidance is provided it is highly likely to learn features irrelevant to classification. As shown in Figure 1, we set the feature extractor part of ABMIL to the relatively complex ResNet18, which directly takes bags composed of raw CIFAR-100 images as inputs for training. During test, we use the classification accuracy of single image to evaluate instance quality directly. Here ResNet18 (He et al., 2016) was pre-trained on ImageNet (Deng et al., 2009), so the model demonstrated performance above random guess level in the early training epochs. However, as training progressed, the model eventually converged to a completely random guessing algorithm. Additionally, ABMIL maintains a high information entropy for the predicted probability distribution of each instance, indicating the low discriminability and confidence of instance features all over the training process. This suggests that the model has always been learning unintended drifted features. The drifted low-quality features could further impact the pooling strategy during training.

***Selection Drift in pooling stage.*** The goal of the instance pooling stage in ABMILs is to aggregate a representative bag-level feature, thereby eliminating the influence of non-key instances within the bag. ABMIL utilizes attention-based pooling to select key instances and aggregate them into an overall bag-level feature. A general form of attention-based pooling is as follows:

$$H_{bag} = \sum_{i=1}^{n} \overrightarrow{a_i} H_i, \quad \text{where} \quad \overrightarrow{a_i} = \frac{exp(H_i^\top \overrightarrow{w})}{\sum_{j=1}^{n} exp(H_j^\top \overrightarrow{w})}; \tag{1}$$

The pooling stage transforms $H$ into a weight vector consistent with the bag length $n$ using ***weights-transformation parameter*** $\overrightarrow{w} \in \mathbb{R}^d$, and aggregates the feature matrix $H$ into a bag-level feature $H_{bag}$ based on the weights $\overrightarrow{a} \in \mathbb{R}^t$.

*Proposition 1.* **Attention-based pooling updates the parameter $\overrightarrow{w}$ based on whether the real-time inferred class of instance is consistent with the bag-level ground truth.**

$$-\nabla_{\vec{w}} L = H^T J^T H C^T (\vec{y} - \overrightarrow{p_{bag}}) \quad \textit{(layer-wise differentiation)}$$
$$= \sum_{i=1}^{n} \vec{\phi}^T (\vec{z_i} - \overrightarrow{z_{bag}}) \cdot H_i \quad \textit{(decomposition \& recombination)} \qquad (2)$$
$$\text{s.t.} \quad \sum_{j=1}^{k} \vec{\phi_j} = 0; \quad \vec{\phi_Y} > 0; \quad \vec{\phi_{\overline{Y}}} < 0; \quad \vec{\phi} \in \mathbb{R}^k$$

In a $k$-class ABMIL model, the update formula is shown above, where $L$ represents the negative log-likelihood loss, $Y$ denotes its ground-truth bag label (scalar), $\overline{Y}$ denotes any other label in $[0, k-1]$ except for $Y$, $\vec{y}$ is the bag ground-truth one-hot label, $\vec{y} \in \mathbb{R}^k$, $\overrightarrow{p_{bag}}$ expresses the model output prediction, $\overrightarrow{p_{bag}} \in \mathbb{R}^k$, $J$ expresses the Jacobian matrix with shape $\mathbb{R}^{n \times n}$, and $\vec{z}, \overrightarrow{z_{bag}} \in \mathbb{R}^k$ are the logits produced by the model's linear classification layer of the instance feature $H_i$ and the bag-level feature $H_{bag}$, respectively.

In the detailed derivation presented in Appendix A.4, we have already verified the properties of $\vec{\phi}$. Next, we first define $\vec{u} = \vec{z_i} - \overrightarrow{z_{bag}}$. Imagine for an instance $x_i$, the corresponding logits output $\vec{z_i}$ is higher than the overall bag level $\overrightarrow{z_{bag}}$ at position $Y$ and lower at position $\overline{Y}$. Therefore, compared to the overall bag output $\overrightarrow{z_{bag}}$, the logits output of instance $\vec{z_i}$ after the softmax function is closer to the one-hot ground truth vector $\vec{y}$. This is what the model considers as a key instance in this bag. We say that when the instance $x_i$ has such a high level of consistency with the ground truth, the vector $\vec{u}$ satisfies the following constraints. In this sense, the fact and the following constraint are equivalent:

$$\vec{u_Y} > 0; \quad \vec{u_{\overline{Y}}} < 0; \quad \vec{u} \in \mathbb{R}^k. \qquad (3)$$

We define $con(i) = \vec{\phi}^T \vec{u}$, and then we have

$$-\nabla_{\vec{w}} L = \sum_{i=1}^{n} con(i) \cdot H_i. \qquad (4)$$

In the final form above, the sign of the consistency parameter $con(i)$ directly indicates the update direction of $\vec{w}$. It is easy to verify that when $\vec{u}$ satisfies the above constraints, implying that the instance has a high level of consistency with ground truth (key instance), the consistency parameter $con(i)$ is positive definite. In this case, the weights-transformation parameter $\vec{w}$ is updated in the positive direction of $\vec{H_i}$, which will further lead to an increase in the corresponding weight of the key instance $x_i$. Conversely, the opposite outcome occurs. Thus far, we could summarize the attention mechanism as a complete statement in Proposition 1 above.

As a selection process, the intention of attention-based pooling is to gradually assign higher/lower weights to key/non-key instances during the learning process. Based on the analysis above, the attention-based mechanism involves examining the real-time probabilistic inference results of instances and updating weights $\vec{w}$ in a direction decided by the instances' consistency level with the bag-level ground truth. The transformation parameter $\vec{w}$ is finally updated based on the instance features $H$. Considering when simple data inputs, where the given $H$ is either fixed or varies regularly, the attention mechanism can continuously update regular instance features to the parameter $\vec{w}$ based on consistency, leading to increased/decreased weights of corresponding key/non-key instances. However, when complex data and the feature extractor with great capability are introduced, the continuous drastic variation of $H$ can significantly impact the learning of weight parameters $\vec{w}$. For key instances with inferences more consistent with the ground truth, features are often irregular and constantly changing. Thus, ***an update from key instance to $\vec{w}$ does not mean an increase in the weight of any key instance***. The drift of features further leads to selection drift in this case.

## 3.2 PROBABILITY-SPACE ALIGNMENT: TOWARDS FEATURE DRIFT

In the representation learning stage, one major issue causing feature drift is the lack of effective guidance for instances. Representation learning is typically guided by labels for classification tasks. In MIL, due to the absence of guidance from instance-level labels, the models often lack confident inference for instances, as reflected in the high entropy of instance probability inference during training shown in Figure 1. Data augmentation technique is often used in weakly supervised learning to enhance the robustness of the model(Sheng et al., 2024)(Wu et al., 2022). To provide effective guidance, we propose an objective called probability-space alignment based on pseudo-label inference, with $CE$ representing cross entropy function:

$$\boldsymbol{L}_{ins}(X) = \frac{1}{n \cdot |\mathcal{A}(x)|} \sum_{i=1}^{n} \sum_{x' \in \mathcal{A}(x_i)} CE(\overset{*}{l}{}^{(i)}, \overrightarrow{p}(x')), \tag{5}$$

where $\overset{*}{p}{}_s^{(i)} = \frac{\prod_{x' \in \mathcal{A}(x_i)} \overrightarrow{p_s}(x')}{\sum_{j=1}^{k} \prod_{x' \in \mathcal{A}(x_i)} \overrightarrow{p_j}(x')}$, $\overset{*}{p}{}^{(i)} \in \mathbb{R}^k$, and $\overset{*}{l}{}^{(i)} = onehot(\overset{*}{p}{}^{(i)})$.

*Proposition 2.* **For an augmented set $\{x', x'', ...\} \in \mathcal{A}(x)$, the optimal target pseudo-label $\overset{*}{l}$ is the one-hot encoding of the normalized mutliplicated probabilities $\overset{*}{p}$ within the set.**

**Proof:** We denote the position where the elements in optimal one-hot label $\overset{*}{l}$ equals 1 as $c$, and for **any** non-optimal one-hot label $l$, the position where it equals 1 as $r$. Since $\overset{*}{l}$ is obtained through Proposition 2, the product of probabilities in the augmented set at position $c$ is greater than at position $r$. We have

$$\boldsymbol{L}_{ins}(X; \overset{*}{l}) - \boldsymbol{L}_{ins}(X; l) = \frac{1}{n \cdot |\mathcal{A}(x)|} \sum_{i=1}^{n} \sum_{x' \in \mathcal{A}(x_i)} (CE(\overset{*}{l}{}^{(i)}, \overrightarrow{p}(x')) - CE(l^{(i)}, \overrightarrow{p}(x'))) \tag{6}$$

$$= \frac{1}{n \cdot |\mathcal{A}(x)|} \sum_{i=1}^{n} \sum_{x' \in \mathcal{A}(x_i)} (-\log \overrightarrow{p_c}(x') - (-\log \overrightarrow{p_r}(x'))) \tag{7}$$

$$= \frac{1}{n \cdot |\mathcal{A}(x)|} \sum_{i=1}^{n} \log \frac{\prod_{x' \in \mathcal{A}(x_i)} \overrightarrow{p_r}(x')}{\prod_{x' \in \mathcal{A}(x_i)} \overrightarrow{p_c}(x')} \quad < 0 \tag{8}$$

That is to say, the target label $\overset{*}{l}$ that we adopt in Equation 5 enables an optimal loss for $\boldsymbol{L}_{ins}(X)$.

As shown in Equation 5, for each instance $x$ in the bag, we generate an augmented set $\mathcal{A}(x)$ and aligns the augmented instances in $\mathcal{A}(x)$ to the inferred label $\overset{*}{l}$, which is obtained from Proposition 2.

$$\boldsymbol{L} = \boldsymbol{L}_{bag}(X) + \lambda_T \cdot \boldsymbol{L}_{ins}(X), \quad \text{where } \lambda_T = \begin{cases} \lambda * T, T < \tau \\ \lambda * \tau, otherwise \end{cases}. \tag{9}$$

The overall loss function is presented in Equation 9. Here the term $\boldsymbol{L}_{bag}(X)$ represents the conventional loss of deep MIL models, $\boldsymbol{L}_{bag}(X) = CE(\overrightarrow{y}, \overrightarrow{p_{bag}})$. The coefficient $\lambda_T$ is controlled by the base parameter $\lambda$ and the current training epoch $T$, increasing gradually as the training progresses until it reaches a threshold epoch $\tau$. By default, the base parameter $\lambda$ is set to 0.1.

## 3.3 PROBABILITY-SPACE ATTENTION: TOWARDS SELECTION DRIFT

Probability-space attention applies attention mechanism on the probability space of instances, which is the largest difference compared to traditional attenion-based pooling. Specifically, we introduce the class prototypes $Q$ ($\in \mathbb{R}^{d \times k}$) to achieve the transformation of the representations from the feature space to the probability space. Through real-time estimation based on the class prototypes $Q$ ($\in \mathbb{R}^{d \times k}$), PSMIL turns to select instances in the probability space, achieved by the following formula:

$$H_{bag} = \sum_{i=1}^{n} \overrightarrow{a_i} H_i, \quad \text{where } \overrightarrow{a_i} = \frac{exp(\tilde{P}_i^{\top} \overrightarrow{w})}{\sum_{j=1}^{n} exp(\tilde{P}_j^{\top} \overrightarrow{w})}; \quad \tilde{P} = \Psi(H, Q) \tag{10}$$

Here $\tilde{P}$ ($\in \mathbb{R}^{n \times k}$) expresses a probability estimation matrix, which is obtained by computing the similarity between the instance feature matrix $H$ and the class prototype matrix $Q$, where $\Psi$ expresses the function of similarity estimation. In our implementation, we set $\tilde{P}_i = softmax(H_i^T Q)$. The estimated probability of each instance further participates in attention-based pooling, as shown in Equation 10. In addition, the probability estimation of each instance in $\tilde{P}$ is processed by the $argmax$ function, resulting in label estimation.

Based on the estimated label for each instance, we calculate the corresponding mean value to obtain a new class prototype matrix $Q_{new}$. Specifically, the $j$-th column of $Q_{new}$ is obtained by averaging

the features of the corresponding instances in $H$ with the label estimated as the $j$-th class. The class prototype $Q$ is then updated by a momentum-based strategy at each training step ($t$), formulated as

$$Q^{t+1} = Normalize((1 - \gamma) \cdot Q^t + \gamma \cdot Q_{new}), \tag{11}$$

where $\gamma$ is the hyper-parameter defaulting to 0.001 in our experiments.

Under the same condition in Equation 2, for the weights-transformation parameter $\overrightarrow{w} \in \mathbb{R}^k$, the update rule in probability-space attention would be redefined as follows

$$-\nabla_{\overrightarrow{w}} L = \sum_{i=1}^{n} \overrightarrow{\phi}^T (\overrightarrow{z_i} - \overrightarrow{z_{bag}}) \cdot \tilde{P}_i. \tag{12}$$

Compared with Equation 2, Equation 12 shows that the weights-transformation parameter is updated by the pattern of real-time probability estimation $\tilde{P}$, instead of the real-time feature matrix $H$. From the perspective of the key instance $x_i$, when the inferred result $\overrightarrow{z_i}$ is more consistent with the ground truth $\overrightarrow{y}$ than the bag level result $\overrightarrow{z_{bag}}$, the corresponding estimated probability $\tilde{P}_i$ will also be similar to the ground truth $\overrightarrow{y}$. According to Equation 12, the transformation parameter $\overrightarrow{w}$ is updated in the positive direction close to the ground truth $\overrightarrow{y}$, which ensures increasing the weight of corresponding key instance $x_i$. As an alternative, the probability-space pooling strategy is less susceptible to drastic changes in feature space than the previous attention-based rule.

# 4 EXPERIMENTS AND ANALYSES

## 4.1 SIMULATED EXPERIMENTS TO EVALUATE INSTANCE REPRESENTATION QUALITY

In the context of multi-class multiple-instance learning, current benchmarks only consist of a few simple bag-level evaluation datasets (Briggs et al., 2012; Settles et al., 2007). To evaluate the model's ability in representation learning, we first introduce more simulated datasets for comprehensive evaluation.

Table 1: Statistics of synthesized datasets.

| Dataset | color | #cls | #ins-tr | #ins-te | #dims | #bag-len | #bags-per-cls | #key-percentage(%) |
|---------|-------|------|---------|---------|--------|----------|---------------|--------------------|
| FMNIST | gray | 10 | 60,000 | 10,000 | 28*28 | 64 | 12,00 | 7.8 |
| SVHN | RGB | 10 | 73,257 | 26,032 | 3*28*28 | 64 | 14,59 | 7.8 |
| CIFAR-10 | RGB | 10 | 50,000 | 10,000 | 3*32*32 | 64 | 10,00 | 7.8 |
| CIFAR-100 | RGB | 100 | 50,000 | 10,000 | 3*32*32 | 64 | 100 | 7.8 |

We synthesize the multi-instance bag version based on four corresponding datasets, and the statistics are shown in Table 1. These four datasets all belong to the domain of images, with the difficulty of representation learning increasing sequentially from Fashion-MNIST (denoted as FMNIST) to CIFAR-100 (Xiao et al., 2017; Netzer et al., 2011; Krizhevsky et al., 2009), and we use color, #cls, #ins-tr, #ins-te, #dims, #bag-len, #bag-per-cls, #l-r, and key-percentage to denote the color mode, number of classes, number of training instances, number of testing instances, dimension of instance, bag length, bag counts per class, key instance percentage in a bag in each dataset, respectively. For more detail about synthesized datasets, see Appendix A.3 .

## 4.2 INSTANCE REPRESENTATION QUALITY EVALUATION

**Implementation Details.** All the algorithms including PSMIL are implemented on a single Nvidia RTX 4090 GPU. Training in multi-instance mode on complex data is generally time-consuming, with a single epoch on CIFAR-100 possibly taking over 1.5 hours. We apply the stochastic gradient descent (SGD) optimizer with a momentum of 0.9 and a weight decay of 0.0001. The initial learning rate is chosen from a set of {0.01, 0.001} and is decayed by steps. On first epoch we freeze the backbone as warming up to improve stability. The value of $\lambda$ is selected from a set of {0.1, 0.01}, with the threshold epoch $\tau$ being 10.

**Performance And Analyses.** In Table 2, the difficulty of representation learning increases gradually across four datasets, where a two-layer convolutional neural network(2-CNN) is used as the

Table 2: Accuracies of instance-level image classification reported during 20 epochs.

| Methods | FMNIST | SVHN | CIFAR-10 | CIFAR-100 | CIFAR-10 | CIFAR-100 |
|---|---|---|---|---|---|---|
| ABMIL (Ilse et al., 2018) | 80.81 | 70.58 | Deg | Deg | *69.98 | *46.13 |
| DEMIL (Tang et al., 2024) | 80.48 | 64.39 | Deg | Deg | *69.89 | *44.82 |
| TransMIL (Shao et al., 2021) | 74.14 | 67.37 | Deg | Deg | *55.76 | *Deg |
| DTFDMIL (Zhang et al., 2022) | **81.27** | 78.04 | Deg | Deg | *69.82 | *44.92 |
| PSMIL | 77.62 | **84.38** | **88.59** | **70.4** | *83.66 | *57.31 |

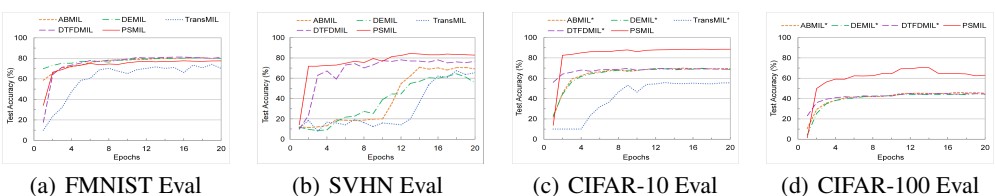

| (a) FMNIST Eval | (b) SVHN Eval | (c) CIFAR-10 Eval | (d) CIFAR-100 Eval |
|---|---|---|---|

Figure 3: Evaluation on four simulated datasets during 20 training epochs.

feature extractor for FMNIST while ResNet18 is applied for the remaining datasets, and the best performance is in bold. "Deg" denotes that the model degraded to random guessing when the feature extractor is involved in training.

From Table 2, it is seen that PSMIL outperforms the compared methods on three more complex datasets (SVHN, CIFAR-10 and CIFAR-100), which indicates that PSMIL is more effective for complex datasets than ABMILs. Meanwhile, the results also demonstrate that most existing models effectively learn discriminative instance features on relatively simple data inputs (FMNIST, SVHN). Particularly on the FMNIST dataset, due to the regularity of the data styles and the simplicity of the feature extractor, applying attention mechanisms in the feature space can also lead to excellent performance. But ABMILs could totally fail for complex datasets, like CIFAR-10 and CIFAR-100. The performance at the level of random guessing and the high entropy of instance prediction probabilities during training (an illustration shown in Figure 1) indicate that these models are not fully capable of handling a general representation learning task, leading to frequent degradation issue during the training process with these complex data.

Moreover, we implement extra experiments which insteads train models in a traditional manner using pre-extracted features for two complex datasets (CIFAR-10 and CIFAR-100). Specifically, we fix the parameters of the pre-trained feature extractor and train the pooling and classification parts of the model. We present the performance marked with "*" in the last two columns of Table 2. In this compromised learning approach, ABMILs demonstrate the ability to learn from fixed features without any degradation issue but still exhibit over a 11% accuracy gap compared with PSMIL on CIFAR-10 and CIFAR-100.

## 4.3 ABLATION STUDY & VISUALIZATION ON REPRESENTATION LEARNING IN MIL

To enhance the representation learning ability for complex data, our idea is to fully transfer ABMILs to probability space. And we propose two strategies: a pooling strategy and an alignment objective strategy.

The detailed ablation of two separate strategies on the existing models is shown in Table 3, where the highest accuracy of each model is reported in 20 training epochs. In Table 3, **PSAtt** and **PSAli** respectively denote the probability-space attention pooling and the incremental probability-space alignment objective, and **RL** is an abbreviation for "representation learning", indicating whether the feature extractor is involved in training. The crossmark (✗) on the option **"RL"** indicates that the model fixes the feature extractor with pre-trained weights on ImageNet, leading to no involvement of representation learning, and the results of ABMIL and DEMIL illustrate that such traditional approach guarantees effective performance. However, when we unfreeze the feature extractor, ABMILs degrade and their results (ABMIL+ and DEMIL+) may "Deg". Noticeably, by applying the **PSAli** objective, the learning ability of ABMILs is reinvigorated, and an accuracy improvement of between 7% and 17% is obtained compared to the original models, as shown as ABMIL++ and DE-

MIL++. Finally, by transferring the pooling strategy to probability space **(PSAtt)**, PSMIL further enhances the quality of instance representations and totally eliminated the degradation issue. We demonstrate the impact of the value of $\lambda$, which controls the power of alignment objective. According to the ablation experiments, in the complex data cases when the model may face a significant risk of degradation, the impact of $\lambda$ is significant. An appropriate parameter $\lambda$ can enhance the stability and performance of the model, preventing the model from failure. We recommend carefully tuning the alignment parameter $\lambda$ on complex datasets. Additionally, the results of a supervised learning manner (ResNet18 (Sup.)) are given in the last row, which shows that our solution PSMIL achieves close supervised learning, with accuracy less than 5%.

Table 3: Ablation results on two challenging datasets CIFAR-10 and CIFAR-100.

| Datasets | CIFAR-10 | | | | CIFAR-100 | | | |
|---|---|---|---|---|---|---|---|---|
| Strategies | **RL** | **PSAli** | **PSAtt** | Acc | **RL** | **PSAli** | **PSAtt** | Acc |
| ABMIL | X | | | 69.98 | X | | | 46.13 |
| DEMIL | X | | | 69.89 | X | | | 44.82 |
| ABMIL+ | ✓ | | | Deg | ✓ | | | Deg |
| DEMIL+ | ✓ | | | Deg | ✓ | | | Deg |
| ABMIL++ ($\lambda_d$) | ✓ | ✓ | | 80.81 | ✓ | ✓ | | 63.72 |
| DEMIL++ ($\lambda_d$) | ✓ | ✓ | | 76.65 | ✓ | ✓ | | 62.13 |
| $\lambda_a : \lambda_T = 0$ | ✓ | | ✓ | 82.79 | ✓ | | ✓ | 64.14 |
| $\lambda_b : \lambda_T = 0.01$ | ✓ | ✓ | ✓ | 83.29 | ✓ | ✓ | ✓ | 64.22 |
| $\lambda_c : \lambda_T = 0.1$ | ✓ | ✓ | ✓ | 84.77 | ✓ | ✓ | ✓ | 66.74 |
| $\lambda_d : \lambda_T = 0.01 * T$ | ✓ | ✓ | ✓ | **88.59** | ✓ | ✓ | ✓ | **70.4** |
| ResNet18(Sup.) | | | | **92.14** | | | | **71.29** |

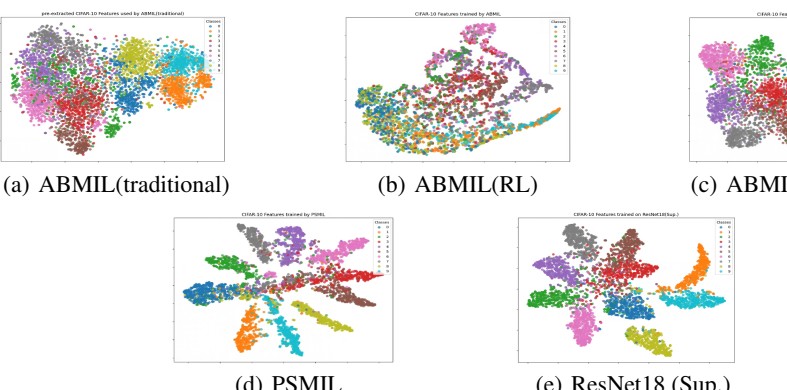

(a) ABMIL(traditional)  (b) ABMIL(RL)  (c) ABMIL(RL+PSAli)

(d) PSMIL  (e) ResNet18 (Sup.)

Figure 4: T-SNE visualization of the image representation on CIFAR-10.

In Figure 4, we further demonstrate the capability of representation learning under different conditions of the ablation experiments. Figure 4(a) represents initial representation distribution provided by pre-trained ResNet18, corresponding to the last second column in Table 2 where ABMIL takes pre-extracted features as input without representation learning. Nonetheless, we cannot simply involve the feature extractor in training, with the degraded features shown in Figure 4(b) . Figure 4(c) demonstrates the enhancement of probability-space alignment strategy in existing MIL models, where ABMIL ensures effective representation learning for complex data through the constraint of probability-space alignment. Figure 4(d) provides the clustering effect of our complete solution PSMIL, where we solve the MIL problem totally in probability space. Comparing Figure 4(b) with 4(d), we can see that PSMIL eliminates the feature drift during representation learning, with similar clustering to a fully supervised level shown in Figure 4(e). See Appendix A.2 for more visualization results.

## 4.4 EVALUATION ON EXISTING BENCHMARKS AND LARGE-SCALE DATASETS

We validated that our model competitive under traditional datasets and conditions. In existing MIL evaluation, many applications might not have on-the-shelf data augmentation methods. This includes datasets such as handcrafted simple benchmarks (Andrews et al., 2002; Dietterich et al., 1997)

and large-scale real-world CAMELYON16,TCGA Lung Cancer datasets(Bejnordi et al., 2017). For implementation details see Appendix A.3.

Table 4: Results on the small benchmark datasets( accuracy ± std-dev). All reimplemented.

| Methods | MUSK1 | MUSK2 | FOX | TIGER | ELEPHANT |
|---|---|---|---|---|---|
| ABMIL(2018) | $0.916 \pm 0.118$ | $0.928 \pm 0.109$ | $0.952 \pm 0.051$ | $0.953 \pm 0.042$ | $0.969 \pm 0.036$ |
| Dual-Stream MIL(2021) | $0.959 \pm 0.053$ | $0.952 \pm 0.066$ | $0.939 \pm 0.060$ | $0.951 \pm 0.053$ | $\mathbf{0.989 \pm 0.023}$ |
| TransMIL(2021) | $0.927 \pm 0.093$ | $0.877 \pm 0.127$ | $0.944 \pm 0.050$ | $0.963 \pm 0.042$ | $0.979 \pm 0.0030$ |
| DEMIL(2024) | $0.963 \pm 0.073$ | $0.961 \pm 0.057$ | $0.941 \pm 0.047$ | $\mathbf{0.965 \pm 0.035}$ | $0.969 \pm 0.034$ |
| RGMIL(2024) | $\mathbf{0.968 \pm 0.060}$ | $0.963 \pm 0.048$ | $\mathbf{0.954 \pm 0.048}$ | $0.949 \pm 0.047$ | $0.965 \pm 0.032$ |
| PSMIL | $\mathbf{0.968 \pm 0.053}$ | $\mathbf{0.966 \pm 0.052}$ | $0.9420 \pm 0.054$ | $0.947 \pm 0.047$ | $0.985 \pm 0.030$ |

Table 5: Results on Large-scale Datasets. Statistics directly collected from Zhang et al. (2022)(Fourkioti et al., 2023)

| Method | CAMELYON16 | | | | TCGA Lung Cancer | | |
|---|---|---|---|---|---|---|---|
| | Multi-scale | ACC(↑) | AUC(↑) | F1(↑) | Multi-scale | ACC(↑) | AUC(↑) |
| ABMIL | Single(20x) | 0.845 | 0.854 | 0.780 | Single(5x) | 0.869 | 0.941 |
| TransMIL | Single(20x) | 0.858 | 0.906 | 0.797 | Single(5x) | 0.883 | 0.949 |
| DTFDMIL(AFS) | Single(20x) | 0.908 | 0.946 | 0.882 | Single(5x) | 0.891 | 0.951 |
| DTFDMIL(MaxMinS) | Single(20x) | 0.899 | 0.941 | 0.865 | Single(5x) | 0.894 | 0.961 |
| DSMIL | Single(20x) | 0.856 | 0.899 | 0.815 | Single(5x) | 0.888 | 0.939 |
| CAMIL | Single(20x) | 0.910 | 0.953 | 0.872 | Single(5x) | 0.916 | 0.975 |
| PSMIL | Single(20x) | **0.922** | **0.956** | **0.921** | Single(5x) | **0.938** | **0.986** |

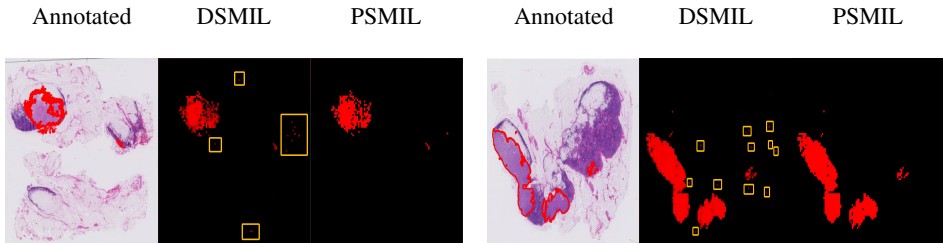

Figure 5: Patch-level Localization of two macro tumors in CAMELYON16

In the traditional pipeline, the features are provided by preprocessing. Therefore, for PSMIL, we only introduced the probability-space attention pooling in all above datasets. Even though these data do not involve complex representation learning our model still achieved SOTA-level performance on the existing datasets as shown in Tabel 4,5. In Table 4 we adopted the untrimmed input data with additional instance label, we also provided the trimmed version in Appendix A.3. And from Figure 5, our solution in probability space could effectively identify the tumor region among the whole slide with significantly less noise than the feature space rule(see yellow boxes), with explicit reference output for each instance(slide patch). For implementation details, see Appendix A.3.

## 5 CONCLUSION

Multiple-instance Learning (MIL) originated in the machine learning era as pure classification, and they typically follow the restriction of fixed or simplified inputs. In current deep MIL models, the representation learning stage is often ignored. In this paper, we provide comprehensive experiments to verify that current deep MIL models tend to learn drifted features in complex tasks, and thus they are still restricted by the complexity of input data. As a countermeasure, we propose a novel network called Probability-Space MIL network (PSMIL). Theoretical analysis and experiments show that we can eliminate drift issues in the feature space by aligning and pooling the instances in the probability space, thereby enhancing the representation learning capability of MIL models for complex tasks. Experiments on public large-scale datasets have also demonstrated the effectiveness of addressing multi-instance problems in the probability space.

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

## A  APPENDIX

### A.1  REPRODUCIBILITY STATEMENT

The codes have been uploaded to Github, and we also presented the important model weights/logs we trained for validation.

## A.2 T-SNE VISUALIZATION OF THE IMAGE REPRESENTATION IN ALL BENCHMARKS

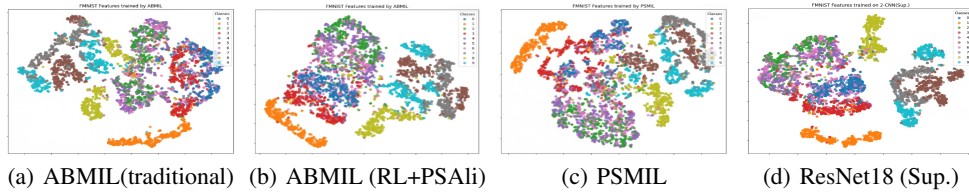

| (a) ABMIL(traditional) | (b) ABMIL (RL+PSAli) | (c) PSMIL | (d) ResNet18 (Sup.) |

Figure 6: FMNIST representation-clustering visualization results.

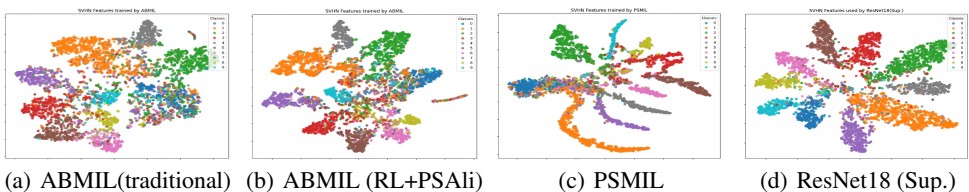

| (a) ABMIL(traditional) | (b) ABMIL (RL+PSAli) | (c) PSMIL | (d) ResNet18 (Sup.) |

Figure 7: SVHN representation-clustering visualization results.

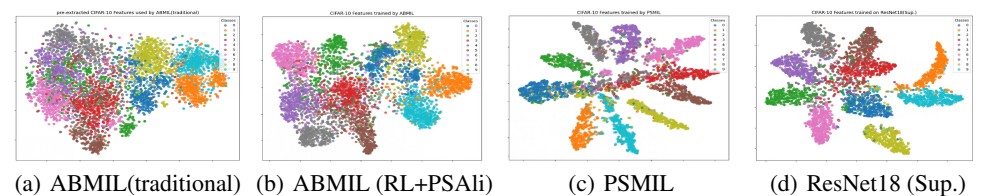

| (a) ABMIL(traditional) | (b) ABMIL (RL+PSAli) | (c) PSMIL | (d) ResNet18 (Sup.) |

Figure 8: CIFAR-10 representation-clustering visualization results.

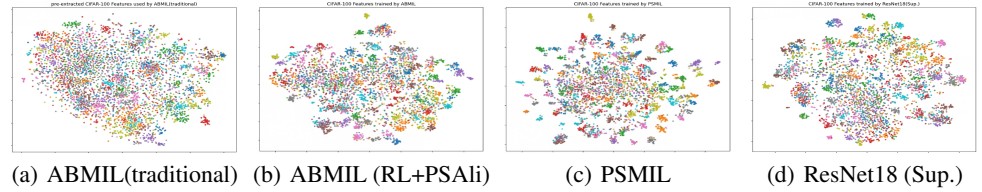

| (a) ABMIL(traditional) | (b) ABMIL (RL+PSAli) | (c) PSMIL | (d) ResNet18 (Sup.) |

Figure 9: CIFAR-100 representation-clustering visualization results.

As demonstrated below, transitioning the MIL solution entirely into the probability space presented more positive impact on rather complex data.

## A.3 DATASETS AND IMPLEMENTATION DETAILS

Detailed properties of 5 benchmark datasets used in Table 4:

| DATASET | MUSK1 | MUSK2 | Elephant | Fox | Tiger |
|---|---|---|---|---|---|
| dimension | 167 | 167 | 231 | 231 | 231 |
| # of bags | 92 | 102 | 200 | 200 | 200 |
| # of positive bags | 47 | 39 | 100 | 100 | 100 |
| # of instances | 476 | 6598 | 1391 | 1320 | 1220 |
| max bag size | 40 | 1044 | 13 | 13 | 13 |
| min bag size | 2 | 1 | 2 | 2 | 1 |

In Table 4,the algorithms are all deep MIL models, which we re-implemented and ran using the same evaluation codes, with replication results reported. In the implementation process, we introduced

two fully connected layers (with ReLU activation function) for all deep MIL models to fine-tune the original inputs.

We borrowed the publicly available evaluation code of RGMIL on Github to re-implement all the methods listed in Table 4 . To be clear, this evaluation code uses untrimmed raw input data, where each instance has an additional feature dimension that includes the instance label. A more common implementation of the evaluation may trim off the instance label from the features. As a result, all methods perform relatively better to different degree in Table 4 than implemented in the trimmed inputs. Generally, considering that the mining of these instance label features is still meaningful, the performance comparison we presented here is similar to the trimmed input version, and this implementation version is also supplemented in our Github repo. Metrics see Table below.

Table 6: Performance comparison provided by Kazeminia et al. (2023) under trimmed input version.

| Method | MUSK1 | MUSK2 | FOX | TIGER | ELEPHANT |
|---|---|---|---|---|---|
| ABMIL | 0.892±0.040 | 0.858±0.048 | 0.615±0.043 | 0.839±0.022 | 0.868±0.022 |
| Gated-ABMIL | 0.900±0.050 | 0.863±0.042 | 0.603±0.029 | 0.845±0.018 | 0.857±0.027 |
| DPMIL | 0.907±0.036 | 0.926±0.043 | 0.655±0.052 | 0.897±0.028 | 0.894±0.030 |
| DSMIL | 0.932±0.023 | 0.930±0.020 | 0.729±0.018 | 0.869±0.008 | 0.925±0.007 |
| BDRMIL | 0.926±0.079 | 0.905±0.092 | 0.629±0.110 | 0.869±0.066 | 0.908±0.054 |
| RGMIL | 0.940±0.070 | 0.920±0.106 | 0.714±0.107 | 0.842±0.088 | 0.915±0.042 |
| TR-RGMIL | 0.946±0.078 | 0.970±0.042 | 0.747±0.054 | 0.961±0.040 | 0.941±0.054 |
| PSMIL | 0.962±0.065 | 0.964±0.057 | 0.734±0.136 | 0.884±0.061 | 0.918±0.052 |

In the new synthesized datasets, training set instances are reorganized into the form of multi-instance bags for training. Instances with class label 0 serve as background instances, randomly sampled within each multi-instance bag as non-key instances. Each multi-instance bag has a fixed length of 64, with 5 being key instances, accounting for 7.8%. The number of categories of bags generated are the same as the number of classes in each original dataset. For example, in CIFAR-100, a bag labeled 78 contains 5 image instances labeled as 78 (maple-tree images), with the remaining 59 non-key instances sampled from instances with label 0. Key instances are not subject to resampling randomly, meaning the same non-zero image will not appear in different bags. During testing, we directly evaluate instance representation quality based on the classification accuracy of single instance images from the original CIFAR test set. To avoid overfitting, we set the bags to be relatively many, with the minimum number of bags per class being 100.

CAMELYON16 is a significant publicly available Whole Slide Image (WSI) dataset for lymph node classification and metastasis detection. It includes 270 training and 129 test slides from two medical centers, all meticulously annotated by pathologists. The TCGA Lung Cancer dataset comprises two non-small cell lung cancer subtypes, LUAD and LUSC, with 1053 slides, including 512 LUSC and 541 LUAD. 10 low-quality LUAD slides are discarded. Unlike CAMELYON16, it lacks patch level annotations and independent test set.

In our experiment, we followed a standard evaluation scheme "5-fold-cv-standalone-test" by DSMIL with as shown in the logs in supplementary detail and codes on Github. Specifically, for the above two medical tasks, in each fold, the best model and corresponding threshold are saved. After the 5-fold cross-validation, 5 best models are obtained which are used to perform inference on the unseen test set. A final prediction for a test sample is the majority vote of the 5 models.

### A.4    DERIVATION OF PROPOSITION 1.

We provide the derivation of the Proposition 1, which explains how real-time instance predictions and bag predictions impact the optimization process of $\overrightarrow{w}$.

**Forward propagation Process:** For a single bag with $n$ instances, $\overrightarrow{y}$ denotes its one-hot encoded label, $Y$ denotes its ground-truth bag label(scalar), $\overline{Y}$ denotes any other label in $[0, k-1]$ except for

$Y$, using the notations from the primary text:

$$\vec{a} = \text{softmax}(H\vec{w});$$
$$H_{bag} = H^T\vec{a};$$
$$\overrightarrow{z_{bag}} = CH_{bag} + \vec{b}; \quad \text{(similar notation for instance } i\text{:}\vec{z_i} = CH_i + \vec{b}) \tag{13}$$
$$\overrightarrow{p_{bag}} = \text{softmax}(\overrightarrow{z_{bag}}) \quad \text{(similar notation for instance } i\text{:}\vec{p_i} = \text{softmax}(\vec{z_i}))$$
$$L = -\vec{y}^T \log\text{softmax}(\overrightarrow{z_{bag}}) \quad \text{(Negative Log-Likelihood Loss)}$$

we have the following simplification process:

$$\begin{aligned}
L &= -\vec{y}^T \log\text{softmax}(\overrightarrow{z_{bag}}) \\
&= -\vec{y}^T \log\frac{exp(\overrightarrow{z_{bag}})}{\mathbf{1}^T exp(\overrightarrow{z_{bag}})} \\
&= -\vec{y}^T[\log(exp(\overrightarrow{z_{bag}})) - \mathbf{1}\log(\mathbf{1}^T exp(\overrightarrow{z_{bag}}))] \\
&= -\vec{y}^T\overrightarrow{z_{bag}} + \log(\mathbf{1}^T exp(\overrightarrow{z_{bag}}))
\end{aligned} \tag{14}$$

Where $\mathbf{1}$ denotes a vector of ones, log represents the natural logarithm, $\text{softmax}(\vec{r}) = \frac{exp(\vec{r})}{\mathbf{1}^T exp(\vec{r})}$, and $exp(\vec{r})$ denotes the element-wise exponentiation. According to the the differentiation rules of matrix operations, element-wise functions $\odot$, and other related methodologies. We have:

$$\begin{aligned}
dL &= -\vec{y}^T d\overrightarrow{z_{bag}} + \frac{\mathbf{1}^T(exp(\overrightarrow{z_{bag}}) \odot d(\overrightarrow{z_{bag}}))}{\mathbf{1}^T exp(\overrightarrow{z_{bag}})} \\
&= -\vec{y}^T d\overrightarrow{z_{bag}} + \frac{exp(\overrightarrow{z_{bag}})^T d(\overrightarrow{z_{bag}})}{\mathbf{1}^T exp(\overrightarrow{z_{bag}})} \\
&= -\vec{y}^T d\overrightarrow{z_{bag}} + \text{softmax}(\overrightarrow{z_{bag}})^T d\overrightarrow{z_{bag}} \\
&= (\text{softmax}(\overrightarrow{z_{bag}})^T - \vec{y}^T)d\overrightarrow{z_{bag}} \\
&= (\overrightarrow{p_{bag}} - \vec{y})^T d\overrightarrow{z_{bag}}.
\end{aligned} \tag{15}$$

In the aforementioned derivation process, a number of simplifying techniques are employed, e.g., the differential of constant is zero. Then based on the differentiation rules we have,

$$\begin{aligned}
dL &= (\overrightarrow{p_{bag}} - \vec{y})^T d(CH_{bag} + \vec{b}) \\
&= (\overrightarrow{p_{bag}} - \vec{y})^T d(CH_{bag}) \\
&= (\overrightarrow{p_{bag}} - \vec{y})^T Cd(H_{bag}) \\
&= (\overrightarrow{p_{bag}} - \vec{y})^T Cd(H^T\vec{a}) \\
&= (\overrightarrow{p_{bag}} - \vec{y})^T CH^T d\vec{a} \\
&= (\overrightarrow{p_{bag}} - \vec{y})^T CH^T d(\text{softmax}(H\vec{w})) \\
&= (\overrightarrow{p_{bag}} - \vec{y})^T CH^T Jd(H\vec{w}) \\
&= (\overrightarrow{p_{bag}} - \vec{y})^T CH^T JHd\vec{w}
\end{aligned} \tag{16}$$

Where $J \in \mathbb{R}^{n \times n}$ is the Jacobian matrix of $\vec{a}$ with respect to $H\vec{w}$, and it follows the following computation rules:

$$J_{i,j} = \begin{cases} a_i(1 - a_j) & \text{if } i = j \\ -a_j a_i & \text{if } i \neq j. \end{cases} \tag{17}$$

Considering the relationship between derivatives and differentials $df = \frac{\partial f}{\partial \vec{x}}^T d\vec{x}$, where $f$ represents a scalar function, and $\vec{x}$ is a column vector. we have,

$$\nabla_{\vec{w}} L = [(\overrightarrow{p_{bag}} - \overrightarrow{y})^T C H^T J H]^T$$
$$= H^T J^T H C^T (\overrightarrow{p_{bag}} - \overrightarrow{y}). \tag{18}$$

As a consequence of the previous results, we have,

$$-\nabla_{\vec{w}} L = H^T J^T H C^T (\overrightarrow{y} - \overrightarrow{p_{bag}}). \tag{19}$$

We express $J^T H C^T (\overrightarrow{y} - \overrightarrow{p_{bag}})$ as $\overrightarrow{\beta} \in \mathbb{R}^n$. Thus, we have,

$$-\nabla_{\vec{w}} L = H^T \overrightarrow{\beta}$$
$$= \sum_{i=1}^{n} \beta_i \cdot H_i$$
$$= \sum_{i=1}^{n} [J^T H C^T (\overrightarrow{y} - \overrightarrow{p_{bag}})]_i \cdot H_i. \tag{20}$$

It is clearly that $H C^T (\overrightarrow{y} - \overrightarrow{p_{bag}}) \in \mathbb{R}^t$, then we have,

$$-\nabla_{\vec{w}} L = \sum_{i=1}^{t} (J^T)_i H C^T (\overrightarrow{y} - \overrightarrow{p_{bag}}) \cdot H_i. \tag{21}$$

To further simplify the above negative gradient formula, we first expand $(J^T)_i$ based formula (14),

$$(J^T)_i = [J_{1i}, J_{2i}, ..., J_{ii}, ..., J_{ni}]$$
$$= [-a_i a_1, -a_i a_2, ..., a_i(1 - a_i), ..., -a_i a_n]$$
$$= -a_i [a_1, a_2, ..., a_i - 1, ..., a_n] \tag{22}$$
$$= -a_i [a_1, a_2, ..., a_i, ..., a_n] + [0, 0, ..., a_i, ..., 0]$$
$$= -a_i \overrightarrow{a}^T + [0, 0, ..., a_i, ..., 0]. \quad \text{(The only non-zero element at index } i \text{ is } a_i)$$

Considering that the computational formulas for instance logits $\overrightarrow{z_i}$ and bag-level logits $\overrightarrow{z_{bag}}$, we have,

$$-\nabla_{\vec{w}}L = \sum_{i=1}^{n}[(J^T)_i HC^T](\vec{y} - \overrightarrow{p_{bag}}) \cdot H_i$$

$$= \sum_{i=1}^{n}[(-a_i \vec{a}^T + [0,0,...,a_i,...,0])HC^T](\vec{y} - \overrightarrow{p_{bag}}) \cdot H_i$$

$$= \sum_{i=1}^{n}[-a_i \vec{a}^T HC^T + [0,0,...,a_i,...,0]HC^T](\vec{y} - \overrightarrow{p_{bag}}) \cdot H_i$$

$$= \sum_{i=1}^{n}[-a_i \vec{a}^T HC^T + a_i H_i C^T](\vec{y} - \overrightarrow{p_{bag}}) \cdot H_i$$

$$= \sum_{i=1}^{n}[-a_i(\overrightarrow{z_{bag}} - \vec{b})^T + a_i(\vec{z_i} - \vec{b})^T](\vec{y} - \overrightarrow{p_{bag}}) \cdot H_i \qquad (23)$$

$$= \sum_{i=1}^{n} a_i[(\vec{b} - \overrightarrow{z_{bag}})^T + (\vec{z_i} - \vec{b})^T](\vec{y} - \overrightarrow{p_{bag}}) \cdot H_i$$

$$= \sum_{i=1}^{n} a_i(\vec{z_i}^T - \overrightarrow{z_{bag}}^T)(\vec{y} - \overrightarrow{p_{bag}}) \cdot H_i$$

$$= \sum_{i=1}^{n} a_i(\vec{y} - \overrightarrow{p_{bag}})(\vec{z_i}^T - \overrightarrow{z_{bag}}^T) \cdot H_i$$

$$= \sum_{i=1}^{n} \vec{\phi}^T(\vec{z_i} - \overrightarrow{z_{bag}}) \cdot H_i, \quad \text{where } \vec{\phi} = a_i(\vec{y} - \overrightarrow{p_{bag}})$$

Considering that $\vec{y}$ is a one-hot encoded hard label and $\vec{p}$ is a distribution, we can easily verify that $\vec{\phi}$ satisfies the constraint: $\sum_{j=1}^{k} \vec{\phi_j} = 0; \vec{\phi_Y} > 0; \vec{\phi_{\overline{Y}}} < 0; \vec{\phi} \in \mathbb{R}^k$ .

