# OpenReview forum: "Rethinking Multiple-Instance Learning From Feature Space to Probability Space"
_ICLR.cc/2025/Conference — ICLR 2025 Poster_

### Official Review · Reviewer_v2T9 · 2024-10-29

**Soundness:** 3
**Presentation:** 3
**Contribution:** 2
**Rating:** 8
**Confidence:** 4

**Summary:**

The author contends that deep MIL models suffer from a drift in the representation of instances during training. This drift, which occurs in the feature space, negatively impacts the model's performance. To address this issue, the paper proposes the Probability-Space MIL network. This network introduces probability-space attention pooling and a probability-space alignment objective. By transforming instances from the feature space to the probability space using class prototypes, the network mitigates the effects of feature drift and consequently enhances the overall performance of the MIL model.

**Strengths:**

1-	The problem is described clearly.

2-	The investigated problem is important.

**Weaknesses:**

1-	Limited downstream evaluation.

2-	The PSMIL approach seems similar to other approaches that use additional attention pooling, such as DSMIL [1] and DTFDMIL [2].

3-	 The paper does not discuss the computational complexity and scalability of the proposed approach in detail, which could be a concern for large-scale applications.

4-	Limited Ablation Studies: While the paper includes some ablation studies, more extensive ablations could strengthen the claims.

[1] Bin Li, Yin Li, and KevinWEliceiri. Dual-stream multiple instance learning network for whole slide image classification with self-supervised contrastive learning. In Proceedings of the IEEE/CVF conference on computer vision and pattern recognition, pp. 14318–14328, 2021.

[2] Hongrun Zhang, Yanda Meng, Yitian Zhao, Yihong Qiao, Xiaoyun Yang, Sarah E Coupland, and Yalin Zheng. Dtfd-mil: Double-tier feature distillation multiple instance learning for histopathology whole slide image classification. In Proceedings of the IEEE/CVF conference on computer vision and pattern recognition, pp. 18802–18812, 2022.

**Questions:**

1- Use another evaluation method, such as localization on WSI datasets, and compare it with the previous approaches.

2-Illustrate the key differences between the proposed approach, DSMIL [1] and DTFDMIL [2].

3-The approach uses additional probability space with the class prototype. Thus, it is worth comparing the proposed approach's scalability and complexity against the previous approaches.

4-We encourage the authors to do more ablation studies for the approach and see how that will impact the performance of the proposed approach. For example, manipulate the hyper-parameter in the equation (11).

---

> ### Author Response · Authors · 2024-11-13
>
> [R]Question-1: As you can see, this paper focuses on problem identification and theoretical analysis. Considering the significance of this issue and the oversimplification of benchmarks in current MIL, we have approached it not as an application-oriented paper but rather from the perspective of providing new benchmarks. The problem description, theoretical analysis, and solution design, together with creating these benchmarks, involve quite a time-consuming workload, but we believe this is meaningful to the field.
> In our problem introduction, we have touched upon application areas of ABMIL such as VAD and WSI. However, these datasets are relatively large-scale and they are not our area of expertise; they are merely used to demonstrate the commonality of ABMIL application issues.
> As noted in our paper, we mention the additional training burden introduced by feature learning and data augmentation, and as pointed out in Appendix A.3, the issue that current MIL models can only process one bag per training step. These issues might require higher performance, GPU memory, and more suitable model designs. However, this is a separate issue not aligned with our paper's original intent.
> Therefore, we present it as potential limitations, as shown in Appendix A.3. Nonetheless, addressing these application issues is indeed a practical necessity. Other reviewers have also mentioned validation in VAD, and after thorough discussion, we will decide on the option of application experiments. Even though this was not our original intent, we might aim to provide these experimental results before the deadline; otherwise, they will only be available in the journal version. This pending concern will be further clarified before ddl.
>
> [R]Question-2: The design  of PSMIL originates from addressing the selection drift in original attention pooling caused by dramatic feature variation. We collectively refer to popular models like DSMIL and DTFDMIL as ABMILs because, in our view, they belong to the same family.
>
> Specifically, DSMIL modifies ABMIL into a pooling scheme that runs original attention pooling and max pooling (a conservative strategy that simply selects the instance with the highest score) in parallel. The instance obtained from the max pooling branch is used to compute a set of weights by taking the inner product with each instance. These weights are averaged with the weights from the attention pooling branch to serve as the overall weights.
> In practice, we found that the stability and performance do not necessarily surpass the original attention pooling. The paper was proposed on WSI data, with another major contribution being the introduction of SimCLR, which brought performance improvements.
>
> For DTFDMIL, the authors designed original attention pooling to be executed twice sequentially, making it somewhat similar to a simplified transformer structure. By dividing a bag into several sub-bags, the first layer of attention pooling is applied within the sub-bags, followed by another round of attention pooling between the sub-bags in the second layer.
>
> Overall, these two models along with other mainstream methods mentioned in the paper, are basically reapplications and extensions of the original attention pooling, with the original version of attention pooling still being their core component. Therefore, in the paper, we classify them as ABMILs. We provide a theoretical analysis of the attention mechanism and clearly identifies the critical flaws that current feature space attention mechanisms bring, with PSMIL as solution.While as the experiment shows, ABMILs face similar issues in complex case. This is the essential difference between PSMIL and others.
>
>  [R]Question-3: We agree with your point, as this issue may indeed be a potential limitation. We have mentioned the additional burden in the appendix and experiment section of our paper, which you may have already noticed. Our intent in this theoretically-focused paper is to provide a potential solution strategy for critical issues, offering forward-looking support for integrating feature learning into the MIL framework. Therefore, we did not present complexity analysis as an ablation study within the paper, but rather expect to address it through appropriate engineering or optimization in different scale future applications.
>
> Overall, under the same conditions, the final version, which involves data augmentation and feature learning, takes approximately five times longer (1h15min for one round of training with CIFAR data) than just fine-tuning the classification head (15min, without feature learning and data augmentation). PSMIL  takes  longer on the same data than ABMIL (1h40min and 1h15min, respectively). If a specific method for complexity evaluation is needed, please let us know for further clarification.
>
>  [R]Question-4: We agree with your point. And we'll present more ablations as suggested. In general, the influence of hyper-parameter
> deserve to be claimed explicitly.

---

> > ### Author Response · Authors · 2024-12-03
> > **Appreciate your efforts and time to our research**
> >
> > Dear reviewer,
> >
> > Thanks very much for your valuable suggestions and effort. With the due approaching, may we politely remind if you forgot to update your rating provided you confirmed your comments are addressed?

---

> > > ### Author Response · Authors · 2024-12-04
> > > **Thanks for your update!**
> > >
> > > We sincerely appreciate your feedback, which has enabled us to present a more comprehensive, reliable, and convincing research. Thank you for the high appreciation to our work.

---

### Official Review · Reviewer_DCds · 2024-11-01

**Soundness:** 2
**Presentation:** 2
**Contribution:** 3
**Rating:** 6
**Confidence:** 4

**Summary:**

This paper addresses the issue of drift in instance representation learning within multiple instance learning (MIL) models by introducing Probability-Space Multiple Instance Learning (PSMIL). In PSMIL, the authors present two main strategies: probability-space attention pooling and a probability-space alignment objective. Experimental results show that the proposed approach improves performance on challenging MIL tasks, achieving results closer to supervised learning standards, while maintaining competitive performance on bag-level classification benchmarks.

**Strengths:**

+ The approach of addressing drift issue in instance representation learning through pseudo-label inference-based probability-space alignment appears novel to me.
+ The authors have provided enough visualizations (e.g., Figure 2) along with the ablation study in Table 3 to justify the effectiveness of the proposed components PSAtt, and PSAli.
+ The strong performance of existing techniques like DTFDMIL is well-explained in the context of the FMNIST dataset.

**Weaknesses:**

+ The authors should ensure all symbols used in the equations are clearly defined and explained. For example, in Equation 5, it is unclear how ${p^{^\rightarrow}_s}(x^\prime)$i s derived. Similarly, in Equation 2, the steps from layerwise differentiation to decomposition and recombination require further clarification.
+ My main concern is the evaluation, as PS-MIL is tested only on synthetic datasets without including any challenging, real-world datasets. Furthermore,  in Table 2, PSMIL performs significantly lower on the FMNIST dataset compared to DTFDMIL, which requires further indepth justification.
+ In the ablation study (Table 3), the standalone contribution of PSAtt, independent of PSAli, is unclear. Including results for PSAtt alone would help clarify its individual impact on overall performance.
+ The impact of hyperparameters, such as $\lambda$ in Equation 9, is not explored. A detailed study on the effect of  $\lambda$ would offer insights into the relative importance of $L_{\text{bag}}$ and $L_{\text{ins}}$ on performance.
+ Video anomaly detection is a key application of the MIL approach [1, 2, 3], providing challenging, real-world datasets. The authors should demonstrate the performance of their method on these datasets in comparison with established baselines [1, 2, 3]. This would strengthen the paper, as only simpler synthetic datasets are currently considered, which do not fully represent real-world MIL challenges.
+ The process of pseudo-label inference in Equation 5 is unclear. Providing additional explanation beyond the equations would aid readers in understanding how pseudo-labels are inferred.

**References**
1. Wu et al. "VadCLIP: Adapting Vision-Language Models for Weakly Supervised Video Anomaly Detection". AAAI2024.
2. Tian et al. “Weakly-supervised Video Anomaly Detection with Robust Temporal Feature Magnitude Learning”. ICCV2021
3. Sultani et al. “Real-world Anomaly Detection in Surveillance Videos”. CVPR2018

**Questions:**

Please refer to Weaknesses section

---

> ### Author Response · Authors · 2024-11-13
>
> [R]Question-1: We adopt the format of presenting the formula first, followed by the explanation of symbol definitions. Formulas 2 and 5 are further clarified as follows:
>
> Eq 2: The derivation process can be found in Appendix A.1. Instance-level recombination is considered the core part of this derivation. Appendix A.1 contains a detailed layer-by-layer differentiation process, along with recombination  and the final attribute proof.
>
> Eq 5: As an alignment enhancement strategy, the method prototype originates from $L_{REG}$ in [1]. Such similar strategy is proven effective in Table 3 of [1]. Compared to [1], we consider using the conjunction of results in the set as the optimal label. As a discrete problem, our inference process is actually obtained through taking the difference. Similar to Eq.2, we assume a negative definite result in derivation, attempt to rearrange the terms based on this, and find suitable constraints. A possible operation that can effectively satisfy these constraints in log space are conjunctions.
>
> [R]Question-2: As you can see, this paper focuses on problem identification and theoretical analysis. Basically this work is theoretical right now. Considering the significance of this issue and the oversimplification of benchmarks in current MIL, we have approached it not as an application-oriented paper but rather from the perspective of providing new benchmarks. The problem description, theoretical analysis, and solution design, together with creating these benchmarks, involve quite a time-consuming workload, but we believe this is meaningful to the field. In our problem introduction, we have touched upon application areas of ABMIL such as VAD and WSI. However, these datasets are relatively large-scale and they are not our area of expertise; they are merely used to demonstrate the commonality of ABMIL application issues. As noted in our paper, we mention the additional training burden introduced by feature learning and data augmentation, and as pointed out in Appendix A.3, the issue that current MIL models can only process one bag per training step. These issues might require higher performance, GPU memory, and more suitable model architecture. However, this is a separate issue not aligned with our paper's original intent. Therefore, we present it as potential limitations, as shown in Appendix A.3. Nonetheless, addressing these application issues is indeed a practical necessity. After thorough discussion, we will decide on the option of application experiments. Even though this was not our original intent, we might aim to provide these experimental results before the deadline; otherwise, they will only be available in the journal version. This pending concern will be further clarified before ddl.
>
> on FMNIST compared with DTFDMIL: For DTFDMIL, the authors designed original attention pooling to be executed twice sequentially, making it somewhat similar to a simplified transformer structure. By dividing a bag into several sub-bags, the first layer of attention pooling is applied within the sub-bags, followed by another round of attention pooling between the sub-bags in the second layer. Basically DTFDMIL is a  reapplication and extension of the original ABMIL, with the original version of attention pooling still being their core component. When we consider PSMIL, it's designed to  solve the issue of drift in instance representation learning when complex data inputs. For FMNIST, the difficulty is lowest across the 4 datasets. With the regularity of the data styles and the simplicity of the feature extractor, the feature perbutation that cause selection drift is muchly alleviated. This case does not sufficiently demonstrate the feature/selection drift issue we investigate, therefore newest ABMILs may achieve slightly better performance than PSMIL.
>
> [R]Question-3: We appreciate the concerns regarding PSAtt. We will supplement this with the ablation results.
>
> Let us reiterate here: on simple data, replacing original attention solely with PSAtt does not result in significant performance improvement; their performance is basically similar. PSAtt, when applied independently on existing benchmarks (Section 4.4), demonstrates performance similar to the traditional ABMILs family. In lines 521-526 and 808-813 of the paper, we also mention the characteristics of PSAtt compared to traditional Attention and suggest that both original attention and PSAtt can be used for simple data.
>
> PSAtt is designed to address the issue of selection drift resulted by feature drift in complex learning tasks, as shown in Section 4.3 and line 796. PSAtt is more suitable for complex data, and when used in conjunction with PSAli, it can achieve the best results.
>
> [R]Question-4: We agree with your point. And we'll present more ablations as suggested. In general, the influence of hyper-parameter deserve to be claimed explicitly.
>
> [R]Question-5: Incorporated in R-2, see above.
>
> [R]Question-6: Incorporated in R-1, see above.

---

> > ### Comment · Reviewer_DCds · 2024-12-02
> > **Thanks for the rebuttal**
> >
> > I appreciate the authors' efforts in addressing my concerns regarding the experimentation on real-world, large-scale datasets such as CAMELYON16 and TCGA-Lung Cancer as well as the impact of the hyperparameter $\lambda$. In recognition of their response, I would like to increase the score to 6. However, I still believe it is crucial to consider Video Anomaly Detection (VAD) tasks, such as the UCF-Crime dataset, and include an experimental result to further strengthen the evaluation of the proposed technique in the paper.

---

> > > ### Author Response · Authors · 2024-12-03
> > > **Thanks for your update**
> > >
> > > We provided the large-scale datasets only in WSI mainly due  to time limitation. We will consider experiments in other applications like VAD further.  Thanks for your  recognition and update again.

---

### Official Review · Reviewer_wt3B · 2024-11-02

**Soundness:** 3
**Presentation:** 2
**Contribution:** 2
**Rating:** 6
**Confidence:** 3

**Summary:**

This work observes that the core issues leading to performance degradation in MIL with Attention-based methods stem from selection shift and feature shift. To address these problems, it introduces prototype learning and feature alignment.

**Strengths:**

This work identifies the core issues leading to failures in Attention-based MIL approaches and proposes targeted solutions. It also provides clear theoretical proofs and achieves superior performance.

**Weaknesses:**

The article employs two core technologies to address the aforementioned issues; however, there are aspects of these technologies that require further clarification. Firstly, introducing prototypes to resolve selection bias is a technique that has been mentioned in "Rethinking Multiple Instance Learning for Whole Slide Image Classification: A Good Instance Classifier is All You Need." Secondly, feature alignment appears to be a specific application of contrastive learning (either bringing enhanced image features closer or aligning probability spaces). If this process were placed before the overall model training (e.g., as a pre-training step), would it achieve similar effects?
Additionally, it is necessary to highlight the differences between this approach and traditional contrastive learning methods.

**Questions:**

The authors need to supplement their work by addressing several issues mentioned in the aforementioned weaknesses.

---

> ### Author Response · Authors · 2024-11-13
>
> We appreciate your recognition of our theoretical contributions and the valuable insights.
>
> [R] Weakness-1: Thank you for providing this citation. According to the model diagram, it appears to be a direct adaptation of PiCO in the multi-instance learning field. It's notable that the paper chooses to compute class prototypes to directly estimate instance labels and incorporates them as instance losses in training, opting for average pooling instead of the widely used attention mechanism. This is a meaningful paper because the authors attempt to showcase advantages at the instance level by adapting PiCO. While we acknowledge the influence of PiCO, the influence to us is primarily conceptual, as it validates the effectiveness of data augmentation for weakly supervised learning tasks. Our main distinction lies in our formulation of the working principles of attention mechanisms and why they can lead to selection bias in end-to-end training. In our approach, class prototypes are used to provide an estimate during the pooling stage to bypass the shortcomings of the attention mechanism we proposed, which is why class prototypes are not highlighted as our main contribution. In other words, probability space pooling can be achieved without introducing class prototypes; it is merely one option. Furthermore, in [1], average pooling is a conservative strategy, thus there is no issue of selection bias. Therefore, the starting points of the two approaches are significantly different. Our paper is essentially theoretical, focusing on revealing both the working principles and critical drawbacks of the widely used feature space attention mechanisms in multi-instance learning, and propose a method to solve the deadly issue. Our data augmentation methods and alignment strategies also differ in implementation, which will be further clarified in the Response-3 below.
>
> [R] Weakness-2: Based on our theoretical analysis in Section 3.1, we predict that improving the quality of pre-trained features cannot address the drift problem during training when using attention mechanisms. However, we are willing to address this concern if further experiments are required. Please specify the experimental conditions and settings explicitly, and we will strive to provide more ablation studies.
>
> [R] Weakness-3: We are familiar with traditional contrastive learning, which may involve SimCLR (self-supervised) and SupCon[2] (supervised contrastive). In fact, we are using a variant of the latter, which originates from [3]. Such pseudo-label contrastive augmentation strategy has been validated in [3] to be significantly effective for the ambiguous problem. Additionally, unlike [3], we provide a multi-instance version and attempt to find the optimal labels generated by the augmented set as the optimization target, further providing mathematical proof.
>
> The articles mentioned above will be further cited in our paper.
>
> [1] Rethinking Multiple Instance Learning for Whole Slide Image Classification: A Good Instance Classifier is All You Need. TCSVT 2024
>
> [2] Supervised contrastive learning. NeurIPS 2020
>
> [3] Adaptive integration of partial label learning and negative learning for enhanced noisy label learning. AAAI 2024

---

> ### Author Response · Authors · 2024-12-03
> **Thank you for your feedbacks!**
>
> Thank you very much for your detailed and valuable feedback. The origin of our method stems from the core issue in Multiple Instance Learning (MIL), which is why problem analysis occupies a core section. In our paper,  we stated that the drift problem is originated from traditional feature-space attention mechanism, while we noticed that [1](we haven't read before) takes simple&conservative average pooling instead of attention pooling.  From the implementation technique perspective, we sought to further apply&improve some may-existing effective techniques as a countermeasure towards addressing the issues to bring robustness and improvements to the traditional MIL model, and we believe our  designs are highly original.
>
> Also, based on the feedback from other reviewers, we have also implemented our method on additional large-scale datasets. We are proud to announce that we have achieved state-of-the-art (SOTA) performance on all metrics(ACC,AUC,F1) in the large-scale datasets. The localization shows that probability space solution could effectively identify the tumor region among the whole slide with significantly less noise than the feature space rule.  The additional  performance further validated the effectiveness in large-scale applications and has been quite insightful for our future research.
>
> With approaching the due, we are sincerely thankful to your recognition of our work, and look forward to a more positive rating.
>
> [1] Rethinking Multiple Instance Learning for Whole Slide Image Classification: A Good Instance Classifier is All You Need. TCSVT 2024

---

### Author Response · Authors · 2024-11-15
**Application Experiment Implementation Suggestion Needed**

As suggested, an additional real-world large-scale application could now be considered to further strengthen our theory. After investigation, we find that the data in real-world only containing bag-level tasks are often much smaller, while an application in VAD with instance-level evaluation task may contain 500+GB raw data (UCF-Crime), and an application in WSI with instance-level evaluation task may contain 700+GB raw data(CAMELYON). We're still trying to download the CAMELYON16 dataset right now, but the scale limits our progress in re-implement our method, thus for us it may be unpractical to provide the re-implement results in several days. We welcome any further specific advise on the dataset selection if practical here to strengthen our theory.

---

### Author Response · Authors · 2024-11-24
**An Enhanced Research Presented as Suggested**

We would like to thank the reviewers for their suggestions to enhance our research. In response to the weaknesses and identified shortcomings, we have made the following modifications in the revised version:

*$\textbf{Limited Evaluation, especially in large-scale applications}$: We were able to produce the experimental results from the large-scale datasets CAMELYON16 and TCGA-Lung in the newest version, along with the localization results. We are proud to announce that we have achieved state-of-the-art (SOTA) performance on all metrics(ACC,AUC,F1) in the large-scale datasets. The localization shows that probability space solution could effectively identify the tumor region among the whole slide with significantly less noise than the feature space rule. Corresponding location: Table 5 and Figure 5, line 524-526 are supplemented in section 4.4. Dataset details of MUSK1,MUSK2,FOX,TIGER,ELEPHANT,CAMELYON16,TCGA are moved to Appendix for page limit.  In line 094,231,481-485, we change the "benchmark datasets" to "various existing datasets" due to the newly added large-scale datasets.

*$\textbf{Lack of ablation experiments, especially in hyper-parameter $\lambda$}$: The hyper-parameter values listed as an ablation result have now been incorporated in the main text, where PSAtt corresponds to the case with a hyper-parameter value of 0. Corresponding location: the rows related with $\lambda$ in Table 3, with line 428-432 are supplemented in section 4.3

*$\textbf{Other suggestions}$:We have adopted other suggestions related to presentation, illustrations, and citations. Corresponding location: new citations added in line 268,485. The mistaken alphabet $\textit{k}$ was corrected to $\textit{c}$ in line 282.

We welcome further criticism and communication from everyone, and we hope for your recognition of our work sincerely.

---

> ### Author Response · Authors · 2024-11-27
> **New Supp  Uploaded**
>
> To further increase the review confidence, we now uploaded the newest $\textbf{traininglog.zip}$ regarding to the newly added large-scale application part(WSI analysisis) in the supplementary material. Unzip the file to see the whole training process and latest performance in the last several lines. The performance is even slightly better than the result we presented in the revised paper days ago.

---

> > ### Comment · Reviewer_v2T9 · 2024-11-28
> >
> > Thank you for the detailed responses, clarifications, and additional experiments. Most of my concerns have been addressed.

---

> > > ### Author Response · Authors · 2024-11-28
> > >
> > > Thank you for recognizing our continuous work in addressing these concerns.  With the valuable suggestions, we presented a more complete research, and  showed  more clear and promising results in real applications. We look forward to an updated version of the initial official review/ratings.

---

### Comment · Area_Chair_NqXH · 2024-11-30
**The deadline for Author/Reviewer discussion period is in three days!**

Dear Reviewers,

Thanks again for providing your constructive comments and suggestions. The deadline for the Author/Reviewer discussion period is in three days (December 2). Please make sure to read the authors' responses and follow up with them if you have any additional questions or feedback.

Best,

AC

---

### Meta-Review · Area_Chair_NqXH · 2024-12-21

**Metareview:**

The paper identifies an important underlying issue in multiple instance learning (MIL) caused by perturbations in the feature space during representation learning, which affects the effectiveness of the widely-used attention-based pooling mechanism. To tackle this problem, the authors propose a Probability-Space MIL network (PSMIL) that leverages a self-training alignment strategy to mitigate feature space drift. Experiments on multiple datasets demonstrate improved performance compared to existing MIL models.

**Strengths**
- The paper provides an interesting perspective by exploring the impact of drifted instance representations under a weakly supervised setting, shedding light on performance degradation in MIL.

**Weaknesses**
- The evaluation is not sufficiently comprehensive. Given the claim that the drift problem is more pronounced for complex data instances, a systematic evaluation on challenging real-world datasets (e.g., video anomaly detection) is necessary to demonstrate the broader applicability of the method to common MIL scenarios.
- The model relies on several critical hyperparameters, but their impact is not thoroughly investigated, which limits insight into its robustness and practicality.

The paper provides an interesting insight into challenges faced by MIL models in complex settings and proposes a reasonable solution. Additional results provided during the rebuttal phase helped address some concerns and strengthened the reviewers’ support for the paper.

**Additional Comments On Reviewer Discussion:**

Most reviewers participated in discussions with the authors. The rebuttal included additional results that addressed some of the major concerns raised by the reviewers. As a result, several reviewers raised their scores, leading to a consensus to accept the paper.

---

### Decision · Program_Chairs · 2025-01-22

Accept (Poster)